# HYBRIDNET: A HYBRID NEURAL ARCHITECTURE TO SPEED-UP AUTOREGRESSIVE MODELS

## ABSTRACT

This paper introduces HybridNet, a hybrid neural network to speed-up autoregressive models for raw audio waveform generation. As an example, we propose a hybrid model that combines an autoregressive network named WaveNet and a conventional LSTM model to address speech synthesis. Instead of generating one sample per time-step, the proposed HybridNet generates multiple samples per time-step by exploiting the long-term memory utilization property of LSTMs. In the evaluation, when applied to text-to-speech, HybridNet yields state-of-art performance. HybridNet achieves a 3.83 subjective 5-scale mean opinion score on US English, largely outperforming the same size WaveNet in terms of naturalness and provide 2x speed up at inference.

## 1 INTRODUCTION

Speech synthesis, also known as text-to-speech (TTS) has variety of applications in human-computer interactions, assistive technology and entertainment productions. The traditional TTS system, which is done with complex hand-engineered pipelines, transforms textual features into high temporal resolution waveforms (e.g., 16KHz). Recent work on neural TTS has demonstrated state-of-the-art results (Arık et al., 2017a;b; Wang et al., 2017; Sotelo et al., 2017). In particular, various neural network architectures (e.g., van den Oord et al., 2016; Mehri et al., 2017) have been proposed as neural vocoders for waveform synthesis.

Recurrent neural network (RNN), especially long short-term memory (LSTM) (Hochreiter & Schmidhuber, 1997), is well-suited to address speech synthesis as it can model long-term dependencies in audio data (e.g., Fan et al., 2014; Zen & Sak, 2015; Graves, 2013). RNNs have been successfully applied in many state-of-the-art neural TTS systems (e.g., Wang et al., 2017; Arık et al., 2017b), and were proven to be more effective than the conventional hidden Markov model (HMM)-based synthesizer. However, RNNs are unsuitable for raw waveform generation at high sampling rate (e.g. 16,000 samples per second), because RNNs process each state sequentially and the computation cannot be paralleled over elements of a sequence at training. In practice, RNNs are usually operated on spectral or hand-engineered features of audio (e.g., Wang et al., 2017; Graves, 2015), which have much fewer time steps than the raw waveform. Most recently, SampleRNN (Mehri et al., 2017) was proposed to tackle this difficulty by combining autoregressive multilayer perceptrons and RNNs in a hierarchical architecture, where different RNN modules operate at different clock rates.

Another line of research investigates the convolutional autoregressive model (e.g., WaveNet (van den Oord et al., 2016)) for waveform synthesis, where the computation over different time-step can be fully parallelized during training. WaveNet can be efficiently trained on audio data with tens of thousands of samples per second. In order to model the long-range dependencies in audio data, WaveNet uses dilated convolution (Yu & Koltun, 2016) to increase the receptive fields of output units, and demonstrates very good performance in speech synthesis. Several state-of-the-art neural TTS systems, such as Deep Voice 1 (Arık et al., 2017a) and Deep Voice 2 (Arık et al., 2017b), use WaveNet to synthesize waveform conditioned on acoustic features (e.g., phoneme duration and fundamental frequency). Despite its full parallelism at training, WaveNet poses daunting computational problem at inference due to the autoregressive nature of the model (see Figure 1 for an illustration).

In addition, although the dilated convolution is very effective at increasing receptive fields, the very long-range connections in deep layer could potentially result in high variance in the output sampling

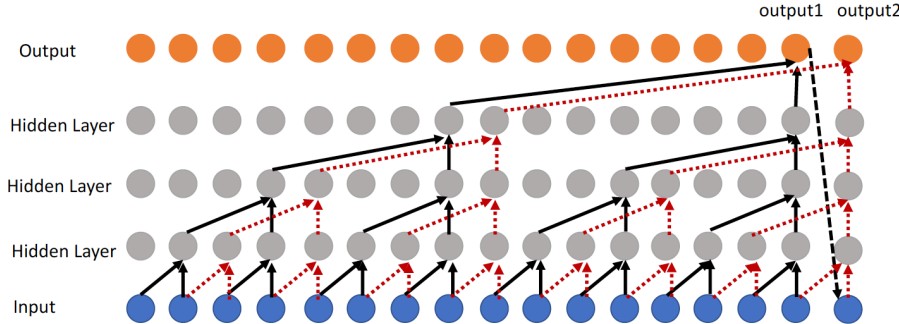

Figure 1: Visualization of a stack of dilated causal convolutional layers with dilations 1, 2, 4, and 8. The autoregressive generation of a single sample depends on the generation of the previous sample at inference time. The sequential dependency largely increases the critical path at inference.

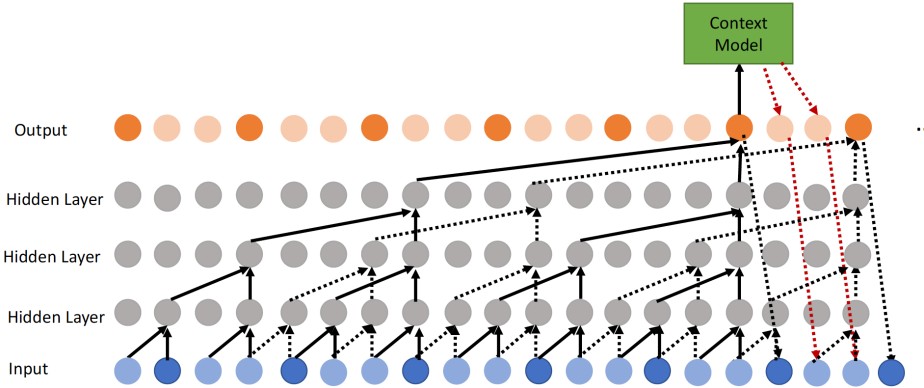

Figure 2: Illustration of HybridNet. It generates more than one sample each time. The light-colored samples are generated by the long-term context model while the dark-colored samples are generated by WaveNet.

distribution. In practice, some buzz noises are commonly observed in the audio samples generated by WaveNet. [1]

In this work, we propose the HybridNet, which combines the strength of convolutional autoregressive model and RNN for audio synthesis. Specifically, our model consists of a WaveNet and a LSTM-based context model, which generate multiple samples simultaneously at each step. As depicted in Figure 2, the context model generates several samples at each step, by taking the output sample of WaveNet as input. As a result, it effectively reduces the sample generation frequency of WaveNet thus reduces inference critical path. Specially, we make the following contributions:

- We propose a novel HybridNet architecture for waveform synthesis. Also, it is worth mentioning that the architecture is generally applicable to speed-up autoregressisve models.

- When we apply HybridNet as a neural vocoder in Deep Voice 2 (Arık et al., 2017b), which is a state-of-the-art neural TTS system, we obtain much higher quality samples compared to the same size WaveNet vocoder according to mean opinion score (MOS) evaluation. In particular, as we increase model layers from 10 to 40, the MOS results of HybridNet improves from 2.84 to 3.83, while the WaveNet vocoder improves from 2.39 to 3.06.

---

[1]Although this problem can be alleviated by some noise-reduction procedure, it can still raise perceptual difference and hurt the naturalness of generated samples.

- Our HybridNet can reduce inference time by half compared to the same layer WaveNet, but only slightly increases the training time. Orthogonal to existing inference time reduction techniques (e.g., caching, optimized gpu kernels), HybridNet can provide additional gains when built on top of existing techniques. [2]

We organize the rest of the paper as follows. Section 2 discusses related work and how this paper is distinct from prior work. In Section 3, we present HybridNet, the hybrid neural vocoder derived from WaveNet. Section 4 presents the experimental results of HybridNet compared to WaveNet in terms of validation error and mean opinion score (MOS).

## 2 RELATED WORK

There have been a number of recent works on building TTS system with neural networks, including WaveNet (van den Oord et al., 2016), SampleRNN (Mehri et al., 2017), Deep Voice 1 (Arık et al., 2017a), Deep Voice 2 (Arık et al., 2017b), Tacotron (Wang et al., 2017), Char2Wav (Sotelo et al., 2017) and VoiceLoop (Taigman et al., 2017). WaveNet and SampleRNN are proposed as neural vocoder models for waveform generation at high temporal resolution. Deep Voice 1 and Deep Voice 2 retain the traditional TTS pipeline, but replace every component (e.g., duration and frequency modeling, waveform synthesis) by neural network based models. In contrast, Tacotron and Char2Wav use sequence-to-sequence architecture (Sutskever et al., 2014; Cho et al., 2014) for building neural TTS systems.

Recent advances in autoregressive models such as WaveNet (van den Oord et al., 2016) and SampleRNN (Mehri et al., 2017) have proven to be successful at modeling short- and medium-term dependencies in raw audio waveform [3], but they rely on external linguistic conditioning or a hierarchical structure to capture long-term dependencies. By using LSTM as an context model, our HybridNet can easily memorize much longer context information across different portions of the audio and generate higher quality samples.

Apart from sample quality, real-time inference is another critical requirement for a production-quality TTS system. The autoregressive models, including WaveNet and SampleRNN, generate one audio sample at each timestep, conditioned on previous generated samples. This sequential dependency between generated samples significantly increases critical path during inference. Various techniques, such as storing the entire model in the processor cache, caching intermediate results in memory, using assembly kernels, are proposed to speed up autoregressive models (Arık et al., 2017a; Ramachandran et al., 2017). In this work, we alleviate the inference bottleneck by designing a new hybrid architecture. Although the resulting speedup is relatively marginal compared to the speedup by investing huge engineering effort (Arık et al., 2017a), it is still an orthogonal technique and can be applied to speed up any autoregressive model. More importantly, it makes real time inference achievable for 40-layers HybridNet, which generates the highest quality samples in our experiment.

## 3 MODEL ARCHITECTURE

In this section, we present the hybrid neural architecture named HybridNet. It keeps the autoregressiveness of WaveNet while using LSTMs to capture long-term context and predict more than one sample at each time-step. The architecture, as shown in Figure 3 consists of two components:

- A WaveNet that takes a time window of raw audio waveform and conditioner as input, [4] and generates a output distribution of the next sample, corresponding to the current time-step.
- A LSTM that takes the sample from WaveNet, hidden state of WaveNet, and conditioners as input to generate next future samples.

---

[2]We noticed that Google has just announced their new state-of-the-art implementation for production (van den Oord et al., 2017), although the technical details haven't been published yet.

[3]When the sampling rate is 16KHz, a 40-layer WaveNet based on 10-layer dilated convolutional block (with dilation 1, 2, 4, ..., 512) can capture dependencies within ~10 milliseconds audio.

[4]We use the same conditioner architecture as in Deep Voice 2 (Arık et al., 2017b)

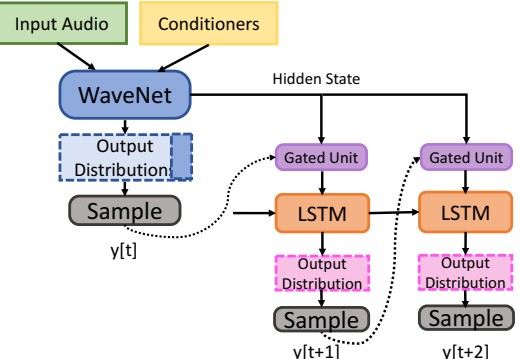

Figure 3: Top-level Architecture of HybridNet: WaveNet produces a window of probability distributions; LSTMs takes a sample for the current time-step from WaveNet output and generates next samples using WaveNet hidden state and conditioners. The generated samples from WaveNet and LSTMs are fed to the input audio.

Similar to WaveNet, HybridNet generates audio samples in an autoregressive fashion by feeding the newly generated samples as the input with the same frequency. Compared to using WaveNet alone, HybridNet reduces inference critical path. Compared to using only LSTMs, HybridNet can generate waveform at high sampling rate.

### 3.1 Architecture of WaveNet

For a given conditioner input $h$, WaveNet models the conditional distribution of the waveform sample $x_t$ based on all previous samples $\{x_1, ..., x_{t-1}\}$. The joint probability of the waveforms is formulated as follows:

$$p(x \mid h) = \prod_{t=1}^{T} p(x_t | x_1, x_2, ..., x_{t-1}, h). \tag{1}$$

In our implementation, WaveNet is conditioned on acoustic features (e.g., phoneme duration and fundamental frequency), which are upsampled by repetition to the same frequency of waveforms.

An n-layer WaveNet consists of an upsampling and conditioning network, followed by $n$ convolution layers with residual connections and gated *tanh* nonlinearity units. Same as Deep Voice (Arık et al., 2017a), the convolution is broken into two matrix multiplications per time-step with $W_{prev}$ and $W_{cur}$. All layers are connected with residual connections with $r$ residual channels. The hidden state of each layer is concatenated to an $n \times r$ matrix and projected to $s$ skip channels. Similar to Deep Voice 2 (Arık et al., 2017b), we remove the $1 \times 1$ convolution between the gated *tanh* nonlinearity and the residual connection, and reuse the conditioner biases for every layer instead of generating a separate bias for every layer.

### 3.2 Multi-sample Prediction with LSTM

The key motivation of HybridNet is to use WaveNet to effectively capture a reasonable range audio context and use LSTM to generate the next few samples more quickly given that context. More specifically, LSTMs generate additional samples based on the sample generated by WaveNet, WaveNet hidden state, and conditioners corresponding to the current time window. We believe that the WaveNet hidden state captures a range of context depending on the size of receptive field, which enables LSTMs to work at a much coarser temporal resolution.

During inference, WaveNet and LSTMs generate samples in a interleaved fashion, where samples generated by LSTMs are fed back to WaveNet as well. The hybrid model effectively reduces the total number of samples generated by WaveNet, thus significantly reduces inference critical path.

Figure 4 depicts the methodology for input embedding and WaveNet hidden state embedding in HybridNet. The model is locally conditioned with acoustic features including frequencies and phoneme duration. Since conditioners have a lower sampling frequency than the audio signal, we first trans-

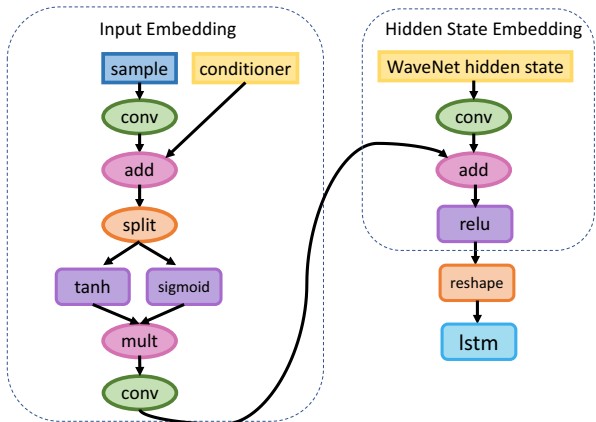

Figure 4: Illustration of Input and WaveNet Hidden State Embeddings for a LSTM Unit. The output sample from WaveNet is conditioned locally with conditioners corresponding to the current time-step. WaveNet hidden state containing long-term context is embedded into the conditioned input.

form the time series of conditioners using a transposed convolutional network. WaveNet takes a window of previously generated audio samples conditioned with a window of conditioners to predicts the next sample. WaveNet output sample is again conditioned locally. In order to leverage the context generated by WaveNet, HybridNet concatenates WaveNet hidden state with the conditioned input and feeds the concatenated embedding to the LSTM unit. The intuition is to use input embedding to capture short-term features and align the features locally, while use hidden state embedding to capture long-term features and dependencies.

### 3.3 TRAINING

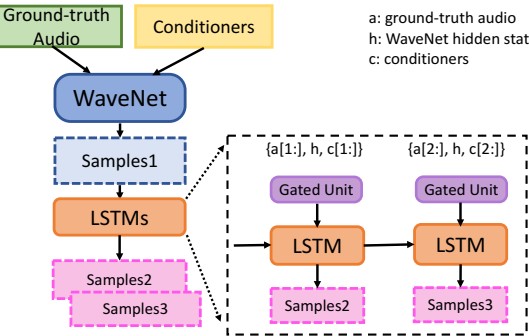

Figure 5: Illustration for Training of a 3-sample Prediction HyrbidNet. Input audio and conditioners are shifted for LSTMs.

We use the ground-truth audios to train the WaveNet component and LSTM component separately. According to our experiment, joint training does not work as well as training separately with ground-truth audios. Similar to WaveNet, HyrbidNet can be trained in parallel with ground-truth audios. However, separate training introduces disparity between the network input in training and in reference. For example, WaveNet takes a sample from HybridNet's output distribution and uses the input for the next time-step during inference. If the sampled output deviates from the ground-truth, WaveNet could get wrong prediction for the next time step. A remedy for this problem will be part of the future work.

During training, we find that the alignment of ground-truth audio and conditioners are critical to the performance of HybridNet. Figure 5 shows that for each prediction made by an LSTM that is n-step ahead of WaveNet's prediction, the ground-truth audio and conditioners need to be shifted by n time-steps ahead.

| Model | Mean Opinion Score (MOS) | Avg. training iteration |
|---|---|---|
| 10-layer WaveNet | $2.39 \pm 0.27$ | 0.225 s |
| 20-layer WaveNet | $2.83 \pm 0.35$ | 0.286 s |
| 40-layer WaveNet | $3.16 \pm 0.30$ | 0.409 s |
| 80-layer WaveNet | $3.06 \pm 0.33$ | 0.624 s |
| 10-layer HybridNet | $2.84 \pm 0.38$ | 0.264 s |
| 20-layer HybridNet | $3.42 \pm 0.34$ | 0.327 s |
| 40-layer HybridNet | $\textbf{3.83} \pm 0.31$ | 0.432 s |
| 80-layer HybridNet | $3.68 \pm 0.31$ | 0.722 s |

Table 1: Mean Opinion Score (MOS) ratings with 95% confidence intervals. We use the crowdMOS toolkit; batches of samples from these models were presented to raters on Mechanical Turk. Since batches contained samples from all models, the experiment naturally induces a comparison between the models.

We train all models with stochastic gradient descent with an Adam optimizer. The baseline WaveNet uses a learning rate of 1e-3 while HybridNet uses a learning rate of 2e-3 with an annealing rate of 0.9886. All the hyperparameters are what we found optimal for each model. Both models are trained for 300k iterations with a batch size of 8.

## 4 EVALUATION

We test vanilla WaveNet and a HybridNet as neural vocoders in Deep Voice 2 system. We train our models on an internal English speech database containing approximately 20 hours of speech data segmented into 13,079 utterances. In our evaluation, we only use single-layer LSTM in the HybridNet but adjust the number of layers of WaveNet component in HybridNet. A single-layer LSTM runs over ten times faster than a WaveNet with more than ten layers. For simplicity, we use n-layer HybridNet short for a HybridNet with n-layer WaveNet component.

HybridNet enables an interesting design space that the inference time can be reduced by either reducing the number of layers in WaveNet or increasing the number of predictions made by LSTMs. We investigated both approaches, and find that a HybridNet with 2-sample prediction has the best performance for a given inference time. As a result, we only evaluate models with varying WaveNet layers in the following sections.

In the following experiments, we evaluate and compare:

1. The naturalness of generated samples as measured by Mean Opinion Score (MOS).

2. The scaling of validation error by changing the number of layers in WaveNet and the WaveNet component in HybridNet.

3. The standard deviation of output distribution as an indication of prediction confidence.

### 4.1 MEAN OPINION SCORE EVALUATION

We evaluate the MOS ratings of WaveNet and HybridNet by varying the number of layers for both models. According to Table 2, a n-layer HybridNet significantly outperforms a n-layer WaveNet in Mean Opinion Score (MOS), but only increases training time marginally (10-20%). In addition, a n-layer HybridNet is 2x faster at inference compared to a n-layer WaveNet. This indicates that HybridNet improves audio synthesis quality and speeds up inference, without hurting training time significantly.

The methodology we used in this work also enables an interesting design space where a variable-layer WaveNet can be combined with an n-step LSTM unit. For example, a 20-layer WaveNet can be combined with a 1-step LSTM unit to match the performance of a 40-layer WaveNet, at a much lower cost (∼4x speedup) in inference time.

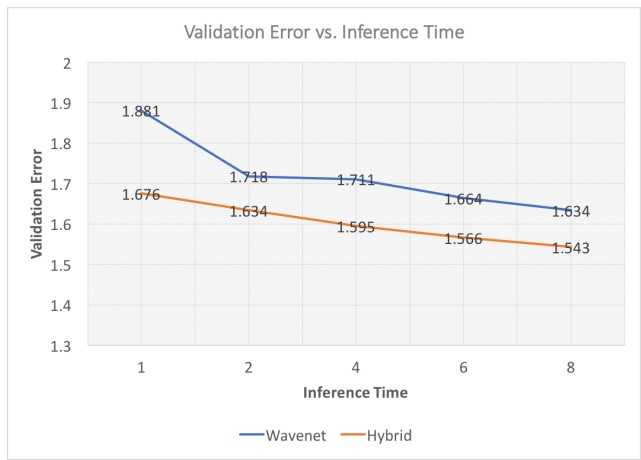

Figure 6: Validation Error vs. Inference Time. Change the number of layers in WaveNet and the WaveNet component in HybridNet changes the inference time and validation error. For a given inference time, HybridNet always outperform WaveNet in terms of validation error.

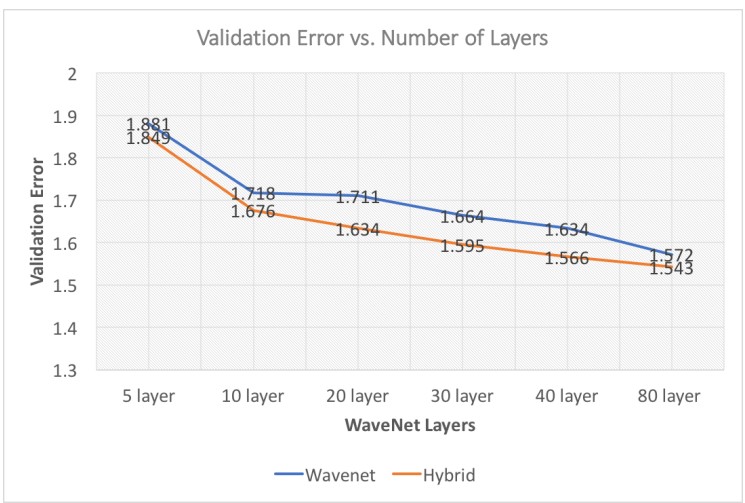

Figure 7: Validation Error vs. Number of Layers in WaveNet. Change the number of layers in WaveNet and the WaveNet component in HybridNet changes validation error. A HybridNet with a n-layer WaveNet component always outperforms a n-layer WaveNet in terms of validation error.

## 4.2 VALIDATION ERROR

In this section, we investigate how validation errors scale with model layers and inference time for WaveNet and HybridNet. Figure 6 shows that increasing the number of layers reduces validation error but also increases inference time. With the same inference time, HybridNet always outperform WaveNet in terms of validation error.

Even without considering inference time, a n-layer HybridNet by itself outperforms a n-layer WaveNet in terms of validation error, as shown in Figure 7. This is probably caused by the reduced variance in the output distribution, where the model is more confident for its prediction. We will demonstrate this point in the following analysis.

| Model | STD |
|---|---|
| 10-layer WaveNet | 0.036 |
| 20-layer WaveNet | 0.036 |
| 40-layer WaveNet | 0.037 |
| 80-layer WaveNet | 0.036 |
| 10-layer HybridNet | 0.032 |
| 20-layer HybridNet | 0.033 |
| 40-layer HybridNet | 0.033 |
| 80-layer HybridNet | 0.034 |

Table 2: Averaged standard deviations (STD) of the output distribution.

## 4.3 STANDARD DEVIATION ANALYSIS

We analyzed the standard deviation of the output distribution for all evaluated models and find HybridNet in general reduces the variance of its output distributions, which indicates "confident" sampling process for generating waveform. This possibly explains why HybridNet generates audios with fewer noises, compared to vanilla WaveNet.

## 5 CONCLUSION

We present a hybrid neural architecture to speed up autoregressive models for audio synthesis. As a demonstration of effectiveness, we design and implement a hybrid model that consists of an autoregressive network named WaveNet and a LSTM model. The hybrid model exploits the long-term memory utilization property and short inference time of LSTMs. We evaluate HybridNet with both Mean Opinion Score (MOS) and validation error, and find that it outperforms WaveNet with the same layer in terms of naturalness and validation error. In addition, HybridNet provides 2x-4x speed-up at inference with comparable generation quality compared to WaveNet. The technique used in HybridNet can be easily applied to another autoregressive model and can be combined with existing inference time reduction techniques.

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
