# OpenReview forum: "HybridNet: A Hybrid Neural Architecture to Speed-up Autoregressive  Models"
_ICLR.cc/2018/Conference — Reject_

### Official Review · AnonReviewer2 · 2017-11-27
**Good results but lacking details about design decisions**

**Rating:** 6
**Confidence:** 5

**Review:**

TL;DR of paper: for sequential prediction, in order to scale up the model size without increasing inference time, use a model that predicts multiple timesteps at once. In this case, use an LSTM on top of a Wavenet for audio synthesis, where the LSTM predicts N steps for every Wavenet forward pass. The main result is being able to train bigger models, by increasing Wavenet depth, without increasing inference time.

The idea is simple and intuitive. I'm interested in seeing how well this approach can generalize to other sequential prediction domains. I suspect that it's easier in the waveform case because neighboring samples are highly correlated. I am surprised by how much an improvement

However, there are a number of important design decisions that are glossed over in the paper. Here are a few that I am wondering about:
* How well do other multi-step decoders do? For example, another natural choice is using transposed convolutions to upsample multiple timesteps. Fully connected layers? How does changing the number of LSTM layers affect performance?
* Why does the Wavenet output a single timestep? Why not just have the multi-step decoder output all the timesteps?
* How much of a boost does the separate training give over joint training? If you used the idea suggested in the previous point, you wouldn't need this separate training scheme.
* How does performance vary over changing the number of steps the multi-step decoder outputs?

The paper also reads like it was hastily written, so please go back and fix the rough edges.

Right now, the paper feels too coupled to the existing Deep Voice 2 system. As a research paper, it is lacking important ablations. I'll be happy to increase my score if more experiments and results are provided.

---

> ### Public Comment · (anonymous) · 2018-01-06
> **We thank the reviewer's great feedback.**
>
> We thank the reviewer's great feedback. In terms of your question:
> * How well do other multi-step decoders do?
> Yes, we have the same question at the early stage of the project. We tried a variety of approaches to generate multiple samples, including a transposed convolution, a vanilla RNN, a high-way, etc. None of them get comparable performance to LSTMs.
>
> * Why does the Wavenet output a single timestep? Why not just have the multi-step decoder output all the timesteps?
> We tried having multi-step decoder to output all timesteps, but unintuitively it is worse than having one sample generated by WaveNet. As pointed out in the result section, LSTM can effectively reduce variance in the output distribution, but this also could reduce the sharpness and naturalness of the audio.
>
> * How much of a boost does the separate training give over joint training? If you used the idea suggested in the previous point, you wouldn't need this separate training scheme.
> The audio quality is substantially better with ground-truth training. Thanks for the suggestion, we will try this idea out.
>
> * How does performance vary over changing the number of steps the multi-step decoder outputs?
> The inference time can be drastically reduced (~2x each time step added) by increasing the number of steps. The audio quality will not degrade noticeably until 6-7 steps (~32-64x speed up) compared to base line.

---

### Official Review · AnonReviewer1 · 2017-11-28
**Right name. Low innovation. Samples please!**

**Rating:** 4
**Confidence:** 5

**Review:**

This paper presents HybridNet, a neural speech (and other audio) synthesis system (vocoder) that combines the popular and effective WaveNet model with an LSTM with the goal of offering a model with faster inference-time audio generation.

Summary: The proposed model, HybridNet is a fairly straightforward variation of WaveNet and thus the paper offers a relatively low novelty. There is also a lack of detail regarding the human judgement experiments that make the significance of the results difficult to interpret.

Low novelty of approach / impact assessment:
The proposed model is based closely on WaveNet, an existing state-of-the-art vocoder model. The proposal here is to extend WaveNet to include an LSTM that will generate samples between WaveNet samples -- thus allowing WaveNet to sample at a lower sample frequency. WaveNet is known for being relatively slow at test-time generation time, thus allowing it to run at a lower sample frequency should decrease generation time. The introduction of a local LSTM is perhaps not a sufficiently significant innovation.

Another issue that lowers the assessment of the likely impact of this paper is that there are already a number of alternative mechanism to deal with the sampling speed of WaveNet. In particular, the cited method of Ramachandran et al (2017) uses caching and other tricks to achieve a speed up of 21 times over WaveNet (compared to the 2-4 times speed up of the proposed method). The authors suggest that these are orthogonal strategies that can be combined, but the combination is not attempted in this paper. There are also other methods such as sampleRNN (Mehri et al. 2017) that are faster than WaveNet at inference time. The authors do not compare to this model.

Inappropriate evaluation:
While the model is motivated by the need to reduce the generation of WaveNet sampling, the evaluation is largely based on the quality of the sampling rather than the speed of sampling. The results are roughly calibrated to demonstrate that HybridNet produces higher quality samples when (roughly) adjusted for sampling time. The more appropriate basis of comparison is to compare sample time as a function of sample quality.

Experiments:
Few details are provided regarding the human judgment experiments with Mechanical Turkers. As a result it is difficulty to assess the appropriateness of the evaluation and therefore the significance of the findings. I would also be much more comfortable with this quality assessment if I was able to hear the samples for myself and compare the quality of the WaveNet samples with HybridNet samples. I will also like to compare the WaveNet samples generated by the authors' implementation with the WaveNet samples posted by van den Oord et al (2017).


Minor comments / questions:

How, specifically, is validation error defined in the experiments?

There are a few language glitches distributed throughout the paper.

---

> ### Public Comment · (anonymous) · 2018-01-06
> **The technique is orthogonal to existing techniques to speedup WaveNet**
>
> We thank the reviewer's feedback but we do feel the hybrid method has its own merits. It is orthogonal to existing techniques including caching.  Even with caching, the critical path caused by dependencies between samples still exists (You cannot generate the next sample earlier). Caching does not fundamentally address this dependency problem. Also caching is subject to hardware. For a different hardware platform (e.g. mobile), there might not be sufficient cache or memory for this purpose.
>
> Mathematically, the hybrid method can be built on top of caching, and still achieve 2x-4x speedup. For instance, caching reduces per sample generation time by k (~20).  The total generation time for a full utterance of n samples would be n*(1/k)*T, where T is the original per sample generation time. With the hybrid method, it can be further reduced to n*(1/k)*T*(1/4).
>
> With respect to the evaluation, we do have a figure of comparison of inference time (Figure 6). We feel it is a fair comparison when we fix the accuracy while comparing the inference time.  And yes, we agree that the key point of the paper is not to improve accuracy,  thus the figures should better convey the key point (reference time).
>
> In terms of the definition of validation error, we partition the training data into 5% validation data and 95% training data and run validation every 250 iterations.  It is not the final test error. Audio quality is measured with MOS as described in the result section.
>
> In terms of audio quality, yes we feel confident to upload samples. The Mechanical Turkers consistently gives better MOS scores for this hybrid model, compared to a WaveNet. Me, personally, listened the samples many times and can confirm that the scores reflect the quality.
> We would love to compare with samples posted by van den Oord et al (2017).

---

### Official Review · AnonReviewer3 · 2017-12-08
**Right choice of problem. Introduces significant independence assumptions.**

**Rating:** 4
**Confidence:** 5

**Review:**

By generating multiple samples at once with the LSTM, the model is introducing some independence assumptions between samples that are from neighbouring windows and are not conditionally independent given the context produced by Wavenet. This reduces significantly the generality of the proposed technique.

Pros:
- Attempting to solve the important problem of speeding up autoregressive generation.
- Clarity of the write-up is OK, although it could use some polishing in some parts.
- The work is in the right direction, but the paucity of results and lack of thoroughness reduces somewhat the work's overall significance.

Cons:
- The proposed technique is not particularly novel and it is not clear whether the technique can be used to get speed-ups beyond 2x - something that is important for real-world deployment of Wavenet.
- The amount of innovation is on the low side, as it involves mostly just fairly minor architectural changes.
- The absolute results are not that great (MOS ~3.8 is not close to the SOTA of 4.4 - 4.5)

---

> ### Public Comment · (anonymous) · 2018-01-06
> **Accuracy is not the main goal of this work. This work is orthogonal to other techniques.**
>
> We really appreciate the reviewer's comments. We also really like Reviewer1's feedback that accuracy is not the main purpose of this paper. We are not trying to outperform SOTA in terms of accuracy but only provide a way to speedup an autoregressive model like WaveNet. We understand that the WaveNet team also have made great progress improving their MOS scores using various techniques (please find their recent paper :) ), but even with those changes, our technique can still be applied to a model that is fundamentally a "WaveNet" and still achieve 2-4x speedup.
>
> Like we explained to Reviewer1, mathematically, the hybrid method can be built on top of other techniques including caching, and still achieve 2x-4x speedup. For instance, caching reduces per sample generation time by k (~20).  The total generation time for a full utterance of n samples would be n*(1/k)*T, where T is the original per sample generation time. With the hybrid method, it can be further reduced to n*(1/k)*T*(1/4).
>
> The speedup can be beyond 2x. The inference time can be drastically reduced (~2x each time step added) by increasing the number of steps produced by the LSTM. The audio quality will not degrade noticeably until 6-7 steps (~32-64x speed up) compared to base line. We would love to add more evaluation in future version.

---

### Decision · Program_Chairs · 2018-01-29
**ICLR 2018 Conference Acceptance Decision**

**Decision:**

Reject

**Comment:**

The paper presents a hybrid architecture which combines WaveNet and LSTM for speeding-up raw audio generation. The novelty of the method is limited, as it’s a simple combination of existing techniques. The practical impact of the approach is rather questionable since the generated audio has significantly lower MOS scores than the state-of-the-art WaveNet model.